# CRISPR with a Double Mismatch Guide RNA Enhances Detection Sensitivity for Low-Frequency Single-Base EGFR Mutation in Circulating Cell-Free DNA of Lung Cancer Patients

**DOI:** 10.3390/cancers17203343

**Published:** 2025-10-16

**Authors:** Kyung Wook Been, Seunghun Kang, Taegeun Bae, Sumin Hong, Garyeong Kim, Junho K. Hur, Woochang Hwang, Boksoon Chang

**Affiliations:** 1Department of Genetics, College of Medicine, Hanyang University, Seoul 04763, Republic of Korea; ysbkw@hanyang.ac.kr (K.W.B.); btg417@naver.com (T.B.); juhur@hanyang.ac.kr (J.K.H.); 2Hanyang Biomedical Research Institute, Hanyang University, Seoul 04763, Republic of Korea; 3Graduate School of Biomedical Science and Engineering, Hanyang University, Seoul 04763, Republic of Korea; rkdekffyd@naver.com (S.K.); sumin1716@naver.com (S.H.); podocyou@hanyang.ac.kr (G.K.); 4Hanyang Institute of Bioscience and Biotechnology, Hanyang University, Seoul 04763, Republic of Korea; 5Department of Pre-Medicine, College of Medicine, Hanyang University, Seoul 04763, Republic of Korea; 6Department of Thoracic and Cardiovascular Surgery, Samsung Medical Center, Seoul 06351, Republic of Korea

**Keywords:** CRISPR, diagnosis, EGFR mutation, cell-free circulating DNA, DNA amplification system

## Abstract

**Simple Summary:**

Many efforts have been made to detect pathogenic mutations in cell-free circulating DNA (cfDNA) derived from blood, which offers a less invasive and more practical alternative to tissue biopsy. However, the low abundance of mutated DNA in blood samples makes detection by high-throughput sequencing challenging. In this study, we developed a highly sensitive CRISPR/Cas12a-based diagnostic system that selectively removes wild-type (WT) DNA to enrich rare mutant DNA, enabling reliable mutation detection through PCR. Applying this approach to cfDNA from 48 non-small-cell lung cancer (NSCLC) patients, we targeted two clinically meaningful EGFR mutations: L858R and exon 19 deletion. Our system successfully detected all 7 L858R-positive cases and 6 of 11 exon 19 deletion-positive cases, consistent with prior tissue biopsy results. These findings demonstrate that our CRISPR/Cas12a-based system enables sensitive and selective detection of low-frequency mutations in cfDNA, supporting its potential use in non-invasive cancer diagnostics.

**Abstract:**

Background/Objectives: Liquid biopsy using cfDNA has emerged as a promising, minimally invasive alternative to traditional tissue biopsy for detecting cancer-associated mutations. However, the extremely low proportion of mutant DNA in cfDNA poses a major challenge for accurate detection, especially when using conventional sequencing methods. To address this limitation, we sought to develop a highly sensitive diagnostic strategy to selectively enrich rare mutant sequences and improve the detection of clinically important mutations in patients with NSCLC. Methods: We established a CRISPR/Cas12a-based diagnostic system designed to selectively cleave WT DNA, thereby increasing the relative abundance of mutant DNA in cfDNA samples. Following Cas12a-mediated WT cleavage, the remaining DNA was subjected to PCR amplification for mutation identification. The system was applied to plasma cfDNA from blood samples of 48 NSCLC patients to evaluate its ability to detect two major EGFR mutations: L858R and exon 19 deletion. Results: The CRISPR/Cas12a-based diagnostic system effectively identified low-frequency EGFR mutations in cfDNA. Specifically, all 7 L858R-positive samples and 6 out of 11 samples harboring exon 19 deletions—previously validated through tissue biopsy—were successfully detected. This demonstrated a high degree of concordance between our liquid biopsy approach and conventional diagnostic methods. Conclusions: Our findings highlight the potential of the CRISPR/Cas12a-based mutation enrichment system as a powerful tool for detecting rare oncogenic mutations in liquid biopsy samples. This technique enhances diagnostic sensitivity and could be broadly applicable for the non-invasive detection of various genetic alterations in cancer and other diseases.

## 1. Introduction

Advances in sequencing technology have revealed a correlation between cancer and DNA mutations, suggesting that DNA mutations can be used as biomarkers for cancer diagnosis [1,2]. Several studies reported that DNA mutations are associated with cancer causes and drug resistance, so the detection of DNA mutations is crucial for cancer therapy [3,4,5].

Recently, growing evidence has demonstrated the potential of diagnosing cancer by detecting DNA mutations from cfDNA, which is released from tumor tissues [6,7,8,9,10,11,12,13,14,15,16,17,18]. In particular, patients with NSCLC which is known to have a higher mortality rate compared to other cancers are required to diagnose in early stage for effective treatment. Given that tyrosine kinase inhibitor, targeted cancer therapy, was shown to be a robust treatment in EGFR mutation-positive NSCLC patients, the early and reliable detection of EGFR mutation as L858R and exon 19 deletion is greatly important for effective treatment, which implies the importance of diagnosis through liquid biopsy [19,20,21]. In this context, analyzing cfDNA obtained from blood samples using liquid biopsy has emerged as a promising diagnostic tool offering a non-invasive rapid and repeatable alternative to conventional tissue biopsy [21]. However, the clinical application of this approach is still hindered by multiple technical and biological obstacles that compromise its diagnostic accuracy. First, the low abundance of cfDNA in blood samples hampers accurate mutation detection, often necessitating additional steps such as PCR [22]. Second, despite significant improvements in sequencing technology, next-generation sequencing (NGS) still exhibits an error rate of approximately 0.5%, making it difficult to distinguish rare mutant alleles from sequencing noise [23]. Furthermore, the overwhelming presence of WT DNA in blood samples significantly complicates the sensitive detection of rare mutant variants.

To overcome these limitations, recent studies have explored the use of the clustered regularly interspaced short palindromic repeats (CRISPR) and CRISPR-associated protein (Cas) system for mutation detection. The CRISPR/Cas system offers programmable and highly specific DNA recognition, making it an attractive tool for selectively targeting and removing WT DNA while preserving rare mutant DNA. This selective cleavage property can be leveraged to improve the signal-to-noise ratio in downstream analyses such as PCR and sequencing [24,25].

Accordingly, we aimed to detect two well-known NSCLC biomarkers—EGFR L858R point mutation and exon 19 deletion—in cfDNA extracted from 48 NSCLC patients. By introducing a deliberate mismatch into the protospacer region of the crRNA, we were able to selectively cleave WT DNA, thereby enriching mutant DNA for downstream PCR amplification. In this study, we demonstrated that the CRISPR-based diagnostic system enables highly sensitive amplification of mutant DNA from liquid biopsy samples of NSCLC patients, highlighting the potential of CRISPR-based diagnostics as powerful tool for early disease detection, personalized treatment planning, and continuous health monitoring.

## 2. Materials and Methods

### 2.1. Purification of LbCas12a Recombinant Protein

Escherichia coli BL21 (DE3) cells were transformed with pET28a–LbCas12a (addgene No. #114070) bacterial expression vectors to express LbCas12a recombinant protein. The cells were cultured in Luria Broth (LB) at 37 °C until the culture reached an O.D of 0.6, and the recombinant protein was induced at 18 °C for 18 h by the addition of isopropyl β-D-thiogalactoside (IPTG, GenDEPOT, Katy, TX, USA) to a final concentration of 1 mM. Thereafter, the cell cultures were centrifuged (4000 rpm, 30 min) to remove the medium, and the pelleted cells were resuspended in lysis buffer (20 mM Tris-HCl (pH 8.0), 300 mM NaCl, 10 mM β-mercaptoethanol (BioRad, Hercules, CA, USA), 1% Triton X-100 (Sigma), and 1 mM phenylmethylsulfonyl fluoride (Sigma, St. Louis, MO, USA)) and lysed by sonication (Qsonica, Newtown, CT, USA). Cell debris was removed by centrifugation at 20,000× *g* for 10 min, and the harvested soluble fraction was mixed with Ni-NTA resin (Takara, Nojihigashi, Japan) for purification. The mixture was incubated with stirring at 4 °C in binding buffer (20 mM Tris-HCl (pH 8.0), 300 nM NaCl) for 1 h. After incubation, the Ni-NTA resins were washed with 10 volumes of the washing buffer. Next, the proteins bound to the Ni-NTA resin were eluted with elution buffer (20 mM Tris-HCl (pH 8.0), 300 nM NaCl, and 200 mM imidazole). Buffer exchange was then conducted on the eluted proteins with storage buffer (200 mM NaCl, 50 mM HEPES (pH 7.5), 1 mM DTT (Sigma, St. Louis, MO, USA), and 40% glycerol (Sigma)) using Centricon filters (Amicon Ultra, Millipore, Burlington, VT, USA). The purity of the recombinant proteins was confirmed by SDS-PAGE (10%, Biorad, Hercules, CA, USA) and Coomassie blue staining (Biorad, Hercules, CA, USA).

### 2.2. In Vitro Transcription of crRNA

For in vitro transcription of crRNA, DNA oligos containing a crRNA, corresponding to target DNA, and a T7 promoter sequence were purchased from COSMO Genentech. The DNA oligo was mixed with T7 RNA polymerase (NEB, Ipswich, MA, USA), 50 mM MgCl2, 100 mM NTPs (ATP, GTP, UTP, CTP), 10× RNA polymerase reaction buffer, Murine RNase inhibitor, 100 mM DTT, and DEPC at 37 °C for 8 h. Then, the mixture was incubated at 37 °C for 1 h with DNase for completely removing of DNA oligo, and the RNA was purified using an RNA purification kit (RBC, New Taipei City, Taipei). the purity and concentration of purified RNA were measured by Nanodrop™ 2000 Spectrophotometer (Thermo-Fisher, Waltham, MA, USA). The purified RNA was aliquoted and stored at −80 °C.

### 2.3. Preparations of PCR Amplicon and In Vitro Cleavage

PCR amplicons inserted L858R and exon 19 deleted mutation were obtained by overlapping PCR using DNA primer including mutation from genomic DNA of HEK293T cells. Purified recombinant Cas12a and crRNA designed to remove DNA other than mutant DNA were premixed and incubated with PCR amplicon at 37 °C for 1 h, followed by at 90 °C for 1 min to inactivate the Cas12a/crRNA ribonucleoprotein complex. Cleavage of PCR amplicon was confirmed by agarose gel electrophoresis (2%).

### 2.4. Cell-Free DNA Purification and Quantification

Circulating DNA was isolated from 1 to 5 mL plasma with the QIAamp Circulating Nucleic Acid Kit (Qiagen, Hilden, Germany). The concentration of purified plasma DNA was determined by qPCR using an 81 bp amplicon on chromosome 1 [26] and a dilution series of intact human genomic DNA (Promega, Madison, WI, USA) as a standard curve. Power SYBR Green was used for qPCR on an HT7900 Real Time PCR machine (Applied Biosystems, Foster City, CA, USA), using standard PCR thermal cycling parameters.

### 2.5. Enrichment of Mutated DNA Through the CRISPR/Cas12a Amplification

To specifically amplify a mutant DNA, purified recombinant LbCas12a protein and designed crRNA were premixed and incubated with PCR amplicons at 37 °C for 1 h, followed by at 90 °C for 1 min to inactivate the Cas12a/crRNA ribonucleoprotein complex. After cleavage of WT DNA, the cleaved mixture was reacted through PCR amplification to amplify a non-cleaved mutant DNA (denaturation at 98 °C for 30 s, primer annealing at 58 °C for 30 s, extension at 72 °C for 30 s, 30 cycles). Enriched PCR products were subjected to a nested PCR step in which barcode sequences were attached (denaturation at 98 °C for 30 s, primer annealing at 58 °C for 30 s, and elongation at 72 °C for 30 s, 35 cycles) to generate target-specific libraries for high-throughput sequencing. Libraries were sequenced on the Illumina iSeq 100 platform (Illumina, San Diego, CA, USA) using paired-end 150 bp reads. Each sample generated approximately 10,000 reads, corresponding to ~10,000× coverage for the targeted regions. Base calling and quality filtering were performed using the instrument’s default pipeline, with >90% of bases typically exceeding a Phred quality score of Q30. Sequencing raw data were analyzed through the CRISPR analyzer web tool to calculate mutant DNA rates [27].

## 3. Result

### 3.1. Establishment of a CRISPR/Cas12a-Based Diagnostic System for Enrichment of Mutated DNA

To specifically amplify mutated DNA, we established an amplification system based on the CRISPR/Cas12a nuclease, which can selectively remove WT DNA. Several studies have shown that Cas12a exhibits superior mismatch discrimination compared to Cas9, which displays reduced specificity when distinguishing WT from mutant DNA with a single nucleotide difference [28,29,30,31,32,33,34,35,36,37,38,39,40,41]. Based on these observations, we adopted LbCas12a, an endonuclease derived from *Lachnospiraceae bacterium*, to enhance the specificity of our system for mutant DNA.

To enable the selective amplification of mutant DNA in a clinical liquid biopsy setting, we designed a workflow in which cfDNA was first extracted from the blood samples of 48 NSCLC patients, followed by PCR amplification of the target regions. The PCR products were then treated with the LbCas12a/crRNA ribonucleoprotein complex, which selectively cleaves WT DNA while leaving mutant alleles intact. The resulting DNA mixture was subjected to an additional round of PCR to further enrich the mutant DNA. This process was iteratively repeated 2–3 times to progressively increase the abundance of low-frequency mutant alleles (Figure 1).

Prior to applying this diagnostic workflow to patient-derived cfDNA samples, we first verified its performance at the in vitro level, confirming that the system could selectively distinguish and amplify mutant DNA in a background of WT DNA.

### 3.2. Selective Enrichment of Mutant DNA Through Mismatch crRNA Targeting

Among various EGFR mutations, L858R is one of the most frequent and clinically significant point mutations observed in NSCLC patients, which serves as an important biomarker for targeted therapy decisions. To evaluate whether LbCas12a can distinguish between the L858R mutation and WT DNA, we first designed a WT crRNA perfectly matched to the WT DNA sequence, which contains a single-nucleotide mismatch when aligned with the L858R mutant DNA within the 20 bp target region. We observed that LbCas12a cleaved both WT and mutant DNA with no significant difference, consistent with its high tolerance to single-nucleotide mismatches (Figure 2a) [38].

To overcome this limitation and improve discrimination, we hypothesized that introducing an intentional mismatch near the L858R site into the crRNA would reduce cleavage efficiency for mutant DNA and thereby enhance selectivity. As illustrated in Figure 2b, a WT crRNA perfectly complementary to the WT DNA allows Cas12a to cleave both WT and mutant L858R sequence (Figure 2b, upper panel). In contrast, when a crRNA harboring a single intentional mismatch proximal to the L858R mutation site is used, the WT DNA still retains only one mismatch and remains susceptible to cleavage (Figure 2b, bottom left panel), whereas the mutant L858R DNA contains two mismatches, which could lead to prevent cleavage (Figure 2b, bottom right panel). This strategy enables the selective degradation of WT DNA and preferential enrichment of the mutant DNA. To further evaluate this strategy, we constructed crRNAs possessing various mismatch combinations at 7 base positions around the L858R point mutation (Figure 2c) and compared the efficiency of amplification of mutant DNA by performing in vitro cleavage using LbCas12a together with designed crRNAs for the DNA mixture that diluted mutant DNA to WT DNA followed by PCR. The crRNA containing a guanine-uracil (dG-rU) mismatch at position 7 achieved the highest enrichment efficiency in subsequent PCR, increasing the quantity of mutant DNA by 3.6-fold (Figure 2c) and exhibited no cleavage activity against mutant DNA in vitro (Figure 2d). Furthermore, previous study reported that adding T-rich tail to the 3′ end of crRNA increased the cleavage efficiency of LbCas12a/crRNA [42]. We added T-rich tailing to the designed crRNA 3′ end, which increased quantity of mutant DNA by 5.9-fold (Figure 2d). Taken together, these results demonstrate that the designed crRNA selectively cleaves WT DNA while leaving the mutant DNA intact, indicating that our system can specifically amplify the mutant DNA. Therefore, we decided to use the crRNA with dG-rU mismatch combination in the position 7 from the PAM motif and T-rich tailing into 3′ end for all subsequent experiments.

### 3.3. High-Sensitivity Detection of EGFR L858R Mutation in NSCLC Liquid Biopsy Samples Using CRISPR/Cas12a-Based Diagnostic System

To investigate the sensitivity of our system using the designed crRNA that selectively amplifies mutant DNA, we performed cleavage reactions on DNA mixtures in which mutant DNA was serially diluted with WT DNA down to a ratio of 1:100,000, followed by PCR amplification. Although L858R mutant DNA was not detected in the 1:100,000 mixture before amplification, the three rounds of amplification increased the L858R mutant DNA rate to 13.8%, which showed that our system could theoretically amplify 0.001% mutations through multiple rounds of CRISPR/Cas12a amplification (Figure 3a).

Building on these findings, we next evaluated the clinical applicability of our CRISPR/Cas12a-based diagnostic system by analyzing cfDNA extracted from the blood of 48 NSCLC patients, including 7 L858R(+) and 41 L858R(–) cases confirmed by tissue biopsy. Prior to applying the system to patient samples, only 1 out of the 7 L858R(+) cases showed detectable mutant DNA (4.9%), while the remaining 6 cases showed no detectable signal. Remarkably, after applying our diagnostic system, L858R mutant DNA was enriched from undetectable levels to as high as 97.4% in these 6 samples (P1, P2, P4, P5, P6, and P7), with sample P6 showing the highest enrichment (Figure 3b).

These results suggested that our CRISPR/Cas12a-based diagnostic system can amplify L858R mutated DNA to the level of sensitivity of tissue biopsy (7/7) while traditional liquid biopsy cannot (1/7). In addition, our system did not detect any mutations in 38 of 41 L858R(−) that were not detected in the tissue biopsy, proving that it is a highly sensitive validation tool with a low false positive.

### 3.4. Extending CRISPR/Cas12a-Based Diagnostic System to EGFR Exon 19 Deletions in NSCLC

To further evaluate the versatility of our system that uses mismatches to discriminate between WT and mutant DNA, we next applied it to the detection of EGFR exon 19 deletion mutations, which represent another major class of alterations in NSCLC. In contrast to the single-nucleotide L858R mutation, exon 19 deletions exhibit diverse deletion sizes, typically ranging from 9 to 18 bp [43,44,45,46,47]. Since Cas12a tends to exhibit lower mismatch tolerance in the context of small deletions, we hypothesized that LbCas12a could distinguish mutant DNA with such deletions from WT DNA without the need for intentional mismatches in the crRNA (Figure 4a). This is because even a crRNA perfectly matched to the WT DNA sequence would inherently form mismatches when bound to a mutant DNA containing deletion, due to the absence of complementary bases in the deleted region. To prove this, we synthesized a DNA fragment containing a 15 bp deletion in the EGFR exon 19 locus and conducted in vitro cleavage assay with LbCas12a and crRNA perfectly matched with WT DNA sequence. As expected, WT crRNA cleaved WT DNA considerably, while mutant DNA containing 15 bp deletion remained uncleaved (Figure 4b). These results suggested that exon 19 WT DNA and mutant DNA can be distinguished without introducing an intentional mismatch to crRNA unlike L858R point mutated DNA. Thus, LbCas12a coupled with WT crRNA can be effectively used for detecting small deletions without the need for mismatch engineering. Next, we investigated whether our system could specifically amplify extremely low levels of EGFR exon 19 deletion mutations. As in the L858R enrichment experiment, DNA mixtures were prepared by serially diluting synthesized 15 bp exon 19 deletion mutant DNA fragments with WT DNA to a ratio of 1:100,000. These mixtures were treated with LbCas12a/crRNA ribonucleoproteins to selectively cleave the WT DNA, followed by PCR amplification to enrich the mutant DNA. While the mutant DNA at 1:100,000 dilution was undetectable by direct PCR amplification without prior CRISPR/Cas12a cleavage, our system successfully enriched it to a detectable level, reaching 12.6% after three rounds of amplification (Figure 4c). To assess the clinical applicability of our method, we applied CRISPR/Cas12a-based enrichment to cfDNA extracted from the blood of 48 NSCLC patients, including 11 exon 19 deletion-positive and 37 deletion-negative cases confirmed by tissue biopsy. Our system enriched exon 19 deletion mutant DNA to as high as 99.1% in 6 of the deletion-positive samples (P1, P4, P8, P9, P10, and P11) (Figure 4d). Notably, in patients P4 and P8, the deletion mutation was present at below 1% prior to amplification, but was enriched to 91.4% and 93.4%, respectively, through CRISPR/Cas12a-based enrichment.

These results demonstrate that our CRISPR/Cas12a-based diagnostic system can be effectively extended to the detection of rare deletion-type mutations, thereby highlighting its versatility in capturing diverse classes of EGFR mutations from clinical liquid biopsy samples.

## 4. Discussion

In the era of precision medicine, there is an increasing demand for diagnostic platforms capable of sensitively and specifically detecting low-frequency mutations from minimally invasive samples such as blood. Liquid biopsy addresses this need by offering a non-invasive alternative to traditional tissue biopsy, enabling real-time monitoring of tumor dynamics and capturing both spatial and temporal heterogeneity. Its ease of sampling and ability to track treatment response and resistance mutations make it a valuable tool in precision oncology. However, despite these advantages, the diagnostic sensitivity of liquid biopsy remains a challenge due to the extremely low abundance of mutant DNA amid a high background of WT DNA in circulation. This limitation is consistent with the detection ranges reported for benchmark cfDNA assays such as droplet digital PCR (ddPCR) and next-generation sequencing with error correction, which can reliably detect mutations down to ~0.01% and ~0.004% allele frequency, respectively [14,22].

Recently, CRISPR-based diagnostic system has shown great promise in enhancing the detection sensitivity and specificity of rare mutations in cfDNA from blood samples, particularly in oncology applications. One notable example is cut-PCR, a CRISPR-mediated mutation enrichment strategy that selectively cleaves wild-type sequences while sparing mutant alleles, thereby enabling highly sensitive downstream detection via PCR. In this approach, Cas9 is guided by a mismatched sgRNA that perfectly matches the wild-type sequence but introduces a single-base mismatch at the mutation site. This design ensures that only wild-type DNA is cleaved, while mutant DNA remains intact and is preferentially amplified. Lee et al. demonstrated that cut-PCR could enrich KRAS G12D mutant alleles by over 80-fold and detect as low as 0.01% mutant allele frequency in a background of wild-type cfDNA [24].

Building on this concept, we aimed to develop a CRISPR-based diagnostic system with improved specificity and broader applicability for detecting clinically relevant mutations in NSCLC. Our approach includes a number of significant innovations that distinguish it from previous systems based on SpCas9-mediated cleavage. First, instead of using SpCas9, our system employs Cas12a, a nuclease with distinct biochemical properties that offers greater mismatch sensitivity and higher specificity in certain contexts [48,49,50]. As the ability to discriminate between mutant and WT DNA is critical for selective amplification, Cas12a proved more effective in enriching mutant alleles. Second, while previous SpCas9-based systems relied on mutations that disrupt the NGG PAM motif to selectively prevent WT DNA cleavage, this PAM-targeted strategy substantially limits the range of applicable mutations. Additionally, SpCas9 is reported to recognize both the canonical NGG PAM and non-canonical motifs such as NGA, which introduces additional considerations in design [51]. In contrast, our Cas12a-based system places point mutations within the protospacer region—the sequence complementary to the guide RNA—rather than within the PAM itself, thereby expanding the number of targetable mutations. Third, to overcome the known issue of Cas12a’s mismatch tolerance—which may hinder its ability to distinguish single-nucleotide variants—we developed a novel strategy to intentionally introduce mismatches into the crRNA near the mutation site. Previous studies have shown that Cas nucleases such as SpCas9, LbCas12a, and engineered Cas12i can produce distinct mutational outcomes when mismatches or indels are present [52,53,54], supporting the rationale for our mismatch-engineered design. Inspired by these findings, we introduced deliberate mismatches to reduce Cas12a’s tolerance and improve allele discrimination. This rational design enhances the differential cleavage efficiency between WT and mutant DNA, thereby improving the diagnostic sensitivity of our assay. Moreover, unlike Cas9-based cut-PCR, our Cas12a system inherently possesses collateral cleavage activity. This feature provides opportunities to directly couple allele-specific enrichment with versatile downstream detection platforms, including fluorescence or lateral flow assays, thereby expanding the diagnostic utility beyond PCR-based readouts. Together, these innovations enable more flexible, sensitive, and mutation-specific amplification from cfDNA, allowing us to clearly identify oncogenic mutations in NSCLC patient samples.

Despite these strengths, our study also revealed certain limitations that warrant careful consideration. For EGFR exon 19 deletions, only 6 of 11 mutation-positive patients were detected. This reduced detection rare likely reflects several contributing factors: (i) the high diversity of exon 19 deletion subtypes with variable breakpoints, whereas our crRNA was designed against the most common subtype; (ii) technical variability in cfDNA quality and yield, influenced by plasma processing, leukocyte contamination, and freeze–thaw history; and (iii) biological heterogeneity in ctDNA shedding depending on tumor burden, metastatic site, and concurrent EGFR-TKI therapy. These factors collectively may account for the lower sensitivity observed. Importantly, our primary aim was to demonstrate the feasibility of mismatch-engineered Cas12a design as a generalizable strategy, rather than to exhaustively cover all exon 19 variants. Nonetheless, future studies with multiplexed crRNA panels and larger patient cohorts will be necessary to comprehensively address this complexity and enhance clinical applicability.

Another important limitation is that our LbCas12a-based platform currently functions as a qualitative assay, providing presence/absence detection of specific EGFR mutations without quantifying allele frequency (AF) or clonality. Since therapeutic decision-making in NSCLC often relies on mutation burden and clonal architecture, this information remains critical for guiding targeted therapy and dosing. We therefore view our assay as complementary to, rather than a substitute for, biopsy-based genetic profiling or quantitative cfDNA approaches such as ddPCR and NGS with error correction. In clinical settings, our system could serve as a rapid pre-screening tool to identify mutation-positive patients, after which quantitative methods would confirm AF and clonality to inform treatment strategy. Future work may further integrate Cas12a-based detection with quantitative platforms, thereby combining the speed and simplicity of CRISPR diagnostics with the quantitative precision of established technologies.

## 5. Conclusions

This study demonstrated that the CRISPR/Cas12a-based diagnostic can specifically amplify 0.001% of the L858R mutant DNA to 13.8% in synthesized DNA mixture. Furthermore, additional iterations of the CRISPR/Cas12a amplification procedure should be able to sufficiently amplify much less mutated DNA than 0.001%. In addition, we demonstrated that CRISPR/Cas12a amplification was able to amplify L858R mutant DNAs in cfDNA of NSCLC patients. All of the L858R positive cfDNA samples verified in tissue biopsy were amplified enough to recognize through our method. Furthermore, CRISPR/Cas12a amplification can be used to enrich mutant DNA with a small insertion or deletion without introducing an intentional mismatch combination as in the L858R mutation. Unlike point mutations, small insertions or deletions do not require additional tolerance reduction because these dramatically reduce mismatch tolerance of Cas12a nuclease. In case of EGFR exon 19 deleted mutation, we demonstrated that Cas12a with crRNA corresponding to deleted DNA sequence can effectively discriminate WT and mutant DNA. However, deleted mutant DNA was amplified in only 6 of 11 cfDNA samples extracted from EGFR exon 19 deletion-positive patient’s blood verified through tissue biopsy, and mutant DNA was not detected in the remaining 5 of 11 cfDNA samples. Given that tumor tissue does not necessarily release a cfDNA, these results showed that mutated DNA might not be released from the tissue even though tumor tissue has a mutation, which makes it difficult to amplify mutant DNA through CRISPR/Cas12a amplification. However, considering our experimental results, the CRISPR/Cas12a amplification system should be a robust validation tool since it can effectively amplify rare mutant DNA that could be confused with sequencing errors. In summary, the CRISPR-based diagnostic system can effectively amplify mutant DNA derived from blood samples and can be utilized for various disease diagnosis.

## Figures and Tables

**Figure 1 cancers-17-03343-f001:**
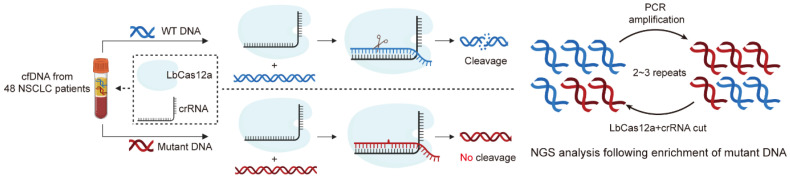
Schematic illustration of the CRISPR/Cas12a-based diagnostic system for enrichment of mutated DNA. cfDNA was extracted from the plasma of lung cancer patients. Extracted cfDNA was reacted with a PCR mixture to amplify the target DNA. To remove specifically the WT DNA, PCR product was reacted with LbCas12a nuclease and crRNA modified a sequence to discriminate between WT and mutated DNA. After cleavage, the PCR reaction was performed to specifically amplify the non-cleaved DNA in reacted mixture. Amplified mutated DNA was attached with a barcoding sequence through nested PCR for high-throughput sequencing.

**Figure 2 cancers-17-03343-f002:**
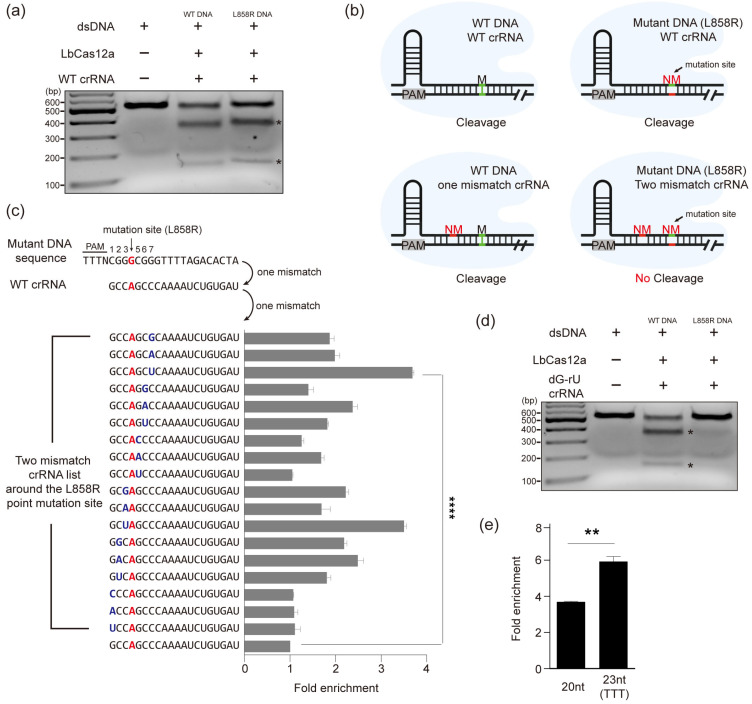
Selective enrichment of mutant DNA through mismatch crRNA targeting. (**a**) In vitro cleavage assay of WT and L858R mutant DNA. The black asterisk indicates cleaved DNA fragments. (**b**) Schematic diagram of differential cleavage of WT and L858R mutant DNA by WT and mismatched crRNA. The light green region indicates WT DNA sequence, and the red region indicates mutated DNA sequence. (**c**) Seven positions around the mutation site were tested with two-mismatch crRNAs. Amplification of DNA mixtures containing mutant DNA was assessed by high-throughput sequencing, and enrichment was calculated from increased L858R frequency. (**d**) In vitro cleavage assay with the optimal crRNA (dG-rU at position 7), which cleaved WT, but not mutant DNA. (**e**) Amplification efficiency was further enhanced by adding a T-tail at the crRNA 3′ terminus. All data are presented as mean ± SED. T-tests were utilized to calculate *p*-values (* *p* = 0.0332, ** *p* = 0.0021, **** *p* = 0.0001).

**Figure 3 cancers-17-03343-f003:**
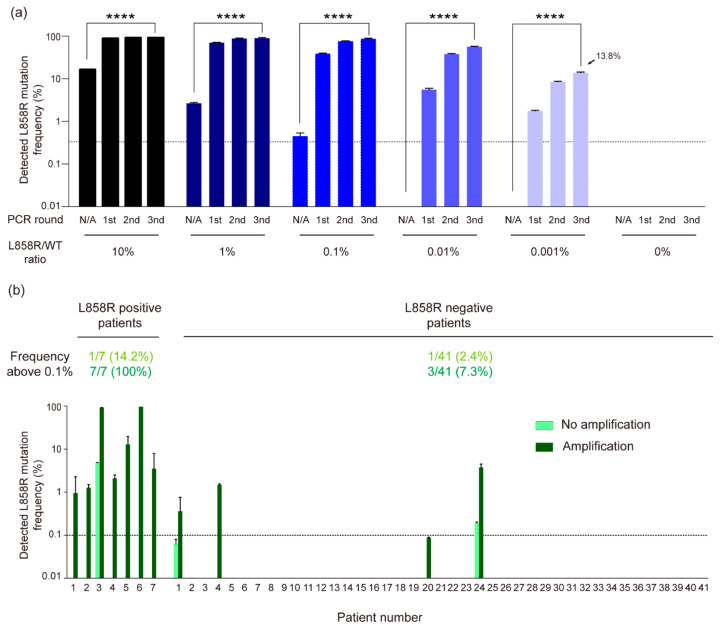
Detection of EGFR L858R mutation using the CRISPR/Cas12a-based diagnostic system in cfDNA. (**a**) Quantitative enrichment analysis of serially diluted L858R mutant DNA (1/100,000 in WT background) by CRISPR/Cas12a amplification. Mutation frequency was measured by high-throughput sequencing; the dotted line indicates the 0.5% sequencing error limit. Data are mean ± SED from two independent experiments (N = 2). Statistical significance was calculated by one-way ANOVA with Tukey’s test (**** *p* = 0.0001). (**b**) Detection of L858R mutation from 48 Lung cancer through CRISPR/Cas12a amplification. 7 clinical samples of the left section were previously verified as L858R mutation-positive clinical samples through tissue biopsy. 41 clinical samples of the right section were previously verified as L858R mutation-negative clinical samples through tissue biopsy. Light green indicates before CRISPR/Cas12a amplification, and dark green indicates after three times of CRISPR/Cas12a amplification. all experiments were conducted at least two times (N = 2). L858R mutation frequency was estimated by high-throughput sequencing. Data are presented as mean ± SED.

**Figure 4 cancers-17-03343-f004:**
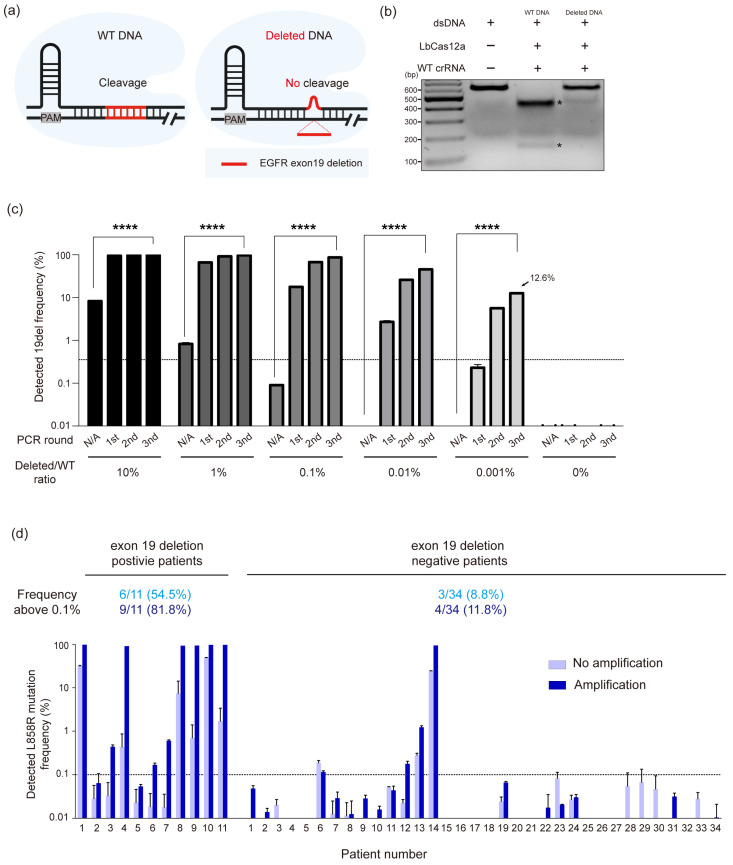
Detection of EGFR Exon19 deleted mutation using the CRISPR/Cas12a-based diagnostic system in cfDNA. (**a**) Schematic showing discrimination of Exon19-deleted DNA from WT DNA by LbCas12a/crRNA ribonucleoprotein. Perfect complementarity to WT DNA enables selective cleavage, as mismatch tolerance is reduced by the long deletion. (**b**) In vitro cleavage assay of WT and deleted DNA using LbCas12a with WT crRNA. Black asterisks indicate cleaved fragments. (**c**) Quantitative enrichment analysis of Exon19-deleted DNA serially diluted in WT DNA (1/100,000) through CRISPR/Cas12a amplification. Mutation frequency was measured by high-throughput sequencing; the dotted line indicates the 0.5% sequencing error limit. Data are mean ± SED from two independent experiments (N = 2). Statistical significance was assessed by one-way ANOVA with Tukey’s test (* *p* = 0.0332, **** *p* = 0.0001). (**d**) Detection of Exon19 deletion in 48 lung cancer cfDNA samples. Eleven biopsy-confirmed positive and 37 biopsy-confirmed negative samples were analyzed before (light blue) and after (dark blue) three rounds of CRISPR/Cas12a amplification. Data are mean ± SED from two independent experiments (N = 2).

## Data Availability

The data that support the findings of this study are available on request from the corresponding author. The data are not publicly available due to privacy or ethical restrictions.

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
