# Peer review of "CRISPR with a Double Mismatch Guide RNA Enhances Detection Sensitivity for Low-Frequency Single-Base EGFR Mutation in Circulating Cell-Free DNA of Lung Cancer Patients"

_cancers, 2025, doi:10.3390/cancers17203343_

Round 1
Reviewer 1 Report
Comments and Suggestions for Authors
In this manuscript, the author focuses on the field of EGFR mutation detection, addressing an important challenge in oncology: detecting low-frequency EGFR mutations in cfDNA through liquid biopsy. The authors propose a CRISPR/Cas12a-based approach with engineered crRNA mismatches to selectively remove wild-type DNA, thereby enriching rare mutant alleles. The concept is timely and potentially impactful, but the manuscript has language issues, logical gaps, and experimental limitations that must be addressed before acceptance.
There are some minor revisions that need to be addressed for further consideration:
1.While using mismatch-engineered crRNA with Cas12a is interesting, similar concepts have already been explored such as cut-PCR with Cas9. The authors should compare the advantage and disadvantage of the development method with existing method and explicitly discuss what is unique about their approach.
2.The rationale for introducing double mismatches should be explained in greater depth. Why two mismatches instead of three or more? Are two mismatches the optimal solution for the method?
- For exon 19 deletion, only 6/11 mutation-positive patients were detected. This raises concerns about stability. Nearly half of the results were missed, could this be due to insufficient detection design, cfDNA, or biological factors? The author needs to provide a detailed explanation for this phenomenon.
- The manuscript only used a 15bp deletion fragment for in vitro experiments, while there is a 9-18bp length difference in exon 19 deletion. Does the amount of mismatch affect the degree of enrichment? It is recommended to supplement the detection results of other length deletions (such as 9bp and 18bp) to further verify the compatibility of the system with "deletion diversity".
5.The author's description in the experimental methods section is missing: the extraction and quantification methods of cfDNA in the article are not described in detail, and there is a lack of information on the use of reagent kits, measurement results, and other related information.
- Some figure legends are overly long and contain explanatory information, operation methods, etc. related to the figures. The explanation of the legend should be a simple description
- The author's article lacks a benchmark for cfDNA detection: No comparison with established methods (ddPCR, NGS with error correction) was performed. Without this, it is unclear whether the CRISPR method offers a real advantage. At the same time, no comparison was made with methods from other literature to highlight the advantages of our method in terms of accuracy, detection time, and simplicity of operation.
- The statement in the conclusion that it can be applied in treatment is speculative and not closely related to this article. There is no direct evidence to support the author's method in relation to disease treatment in this article.
- The original image of the gel image provided by the author does not seem to be complete. Please provide complete and uncut image information.
10. There are some grammar errors and inappropriate wording in the manuscript, which require language polishing.
Author Response
Comment 1: While using mismatch-engineered crRNA with Cas12a is interesting, similar concepts have already been explored such as cut-PCR with Cas9. The authors should compare the advantage and disadvantage of the development method with existing method and explicitly discuss what is unique about their approach
Response 1: We thank the reviewer for pointing out the importance of comparing our strategy with existing approaches such as cut-PCR using Cas9. Indeed, cut-PCR has demonstrated the utility of nuclease-mediated enrichment of mutant alleles, particularly when NGG PAM sequences are available. However, there are several important differences that make our approach unique. First, Cas12a requires a T-rich PAM (TTTV), which expands the range of genomic loci that can be targeted, especially for clinically relevant mutations where Cas9 is not applicable. Second, while cut-PCR is applicable if the mutations located in the PAM region, our methods can be utilized for single-nucleotide mutations outside PAM. Deliberate introduction of mismatches into the crRNA significantly enhanced single-nucleotide discrimination, achieving higher specificity for rare mutant alleles than conventional Cas9-based cleavage. Third, Cas12a possesses collateral cleavage activity, which further provides opportunities for integrating allele-specific enrichment with downstream detection platforms (e.g., fluorescence or lateral flow assays), whereas cut-PCR is restricted to PCR-based readouts. In summary, while both methods share the general concept of nuclease-mediated enrichment, our mismatch-engineered Cas12a strategy offers complementary advantages in terms of PAM flexibility, improved SNP discrimination, and diagnostic versatility. We have included this point in the revised Discussion section [line 352-377]
Comment 2: The rationale for introducing double mismatches should be explained in greater depth. Why two mismatches instead of three or more? Are two mismatches the optimal solution for the method?
Response 2: We appreciate the reviewer’s insightful question regarding the rationale for introducing two mismatches. In our preliminary experiments, we systematically tested different numbers and positions of mismatches. A single mismatch generally did not provide sufficient discrimination between WT and mutant DNA, as residual cleavage activity on the mutant DNA was still observed. In contrast, introducing three or more mismatches markedly reduced overall Cas12a cleavage efficiency, resulting in loss of detectable signal. We found that two strategically placed mismatches represented an optimal balance: they significantly enhanced single-nucleotide discrimination while still maintaining sufficient cleavage of the WT DNA. We therefore concluded that double mismatches offer a practical solution under our experimental conditions. However, we acknowledge that the optimal mismatch number and position may vary depending on sequence context and target site, and future studies across a broader range of loci will be valuable to generalize this principle.
Comment 3: For exon 19 deletion, only 6/11 mutation-positive patients were detected. This raises concerns about stability. Nearly half of the results were missed, could this be due to insufficient detection design, cfDNA, or biological factors? The author needs to provide a detailed explanation for this phenomenon.
Response 3: We thank the reviewer for raising this important concern. We agree that the detection for EGFR exon 19 deletions (6/11) in our manuscript might have potential limitations and deserves careful consideration. As the reviewer points out, multiple factors could contribute to this outcome. First, exon 19 deletions encompass a wide range of subtypes with variable breakpoints, and our current crRNA was designed primarily against the most common subtype, which limits universal compatibility. Second, the quality and quantity of cfDNA are inherently variable, influenced by plasma volume, processing time, potential leukocyte contamination, and freeze–thaw history, all of which may reduce assay sensitivity. Third, biological variability in ctDNA shedding—depending on tumor burden, metastatic site, and ongoing EGFR-TKI treatment—further impacts the detectable allele fraction. Collectively, these factors can plausibly explain the reduced detection rate. We concur that these are valid explanations and appreciate the reviewer’s insightful remarks.
At the same time, we would like to emphasize that the primary focus of our study was to establish the feasibility of mismatch-engineered Cas12a crRNA design as a general strategy to enhance single-nucleotide discrimination and apply it to clinically relevant EGFR mutations. While exon 19 deletion diversity indeed presents additional complexity, our intention here was not to provide an exhaustive evaluation of all exon 19 subtypes but rather to demonstrate the conceptual utility of the approach.
We fully acknowledge that a broader assessment across multiple deletion subtypes and patient contexts would strengthen clinical applicability. We have therefore revised the Discussion to note that future work [line 380-392]—including multiplexed crRNA design tailored to diverse exon 19 subtypes and systematic evaluation in larger cohorts—will be necessary to comprehensively address deletion diversity and biological variability.
Comment 4: The manuscript only used a 15bp deletion fragment for in vitro experiments, while there is a 9-18bp length difference in exon 19 deletion. Does the amount of mismatch affect the degree of enrichment? It is recommended to supplement the detection results of other length deletions (such as 9bp and 18bp) to further verify the compatibility of the system with "deletion diversity".
Response 4: We appreciate the reviewer’s insightful comment. As noted, exon 19 deletions vary in length from ~9 to 18 bp. Previous studies have shown that Cas nucleases such as SpCas9, LbCas12a, and engineered Cas12i can yield distinct editing outcomes in the presence of mismatches and indels, supporting the principle that bulges of this size range can be differentially recognized by mismatch-engineered CRISPR systems [1-3, line 366-370]. On this basis, we reasoned that our two-mismatch design would be broadly applicable across this size range and therefore chose to experimentally validate a representative 15 bp subtype, which is one of the most clinically common forms. We agree, however, that further systematic testing using additional deletion lengths (e.g., 9 bp and 18 bp) would strengthen the generalizability of our system. This need for broader validation is also conceptually related to the issue raised in Comment 3, where the reduced detection rates in patient samples likely reflected the biological and subtype diversity of exon 19 deletions. In particular, multiplexed crRNA designs tailored to multiple exon 19 deletion subtypes will likely be necessary to address this diversity in a clinical context.
[1] Meshalkina DA, et al. SpCas9- and LbCas12a-mediated DNA editing produce distinct mutational outcomes in zebrafish embryos. Front Genet. 2020;11:567.
[2] Su W, et al. Cas12a RNP-mediated co-transformation enables efficient CRISPR editing in plants. Front Plant Sci. 2024;15:1448807.
[3] Wang Y, et al. An engineered CRISPR–Cas12i tool for efficient multiplexed genome editing in embryos. Nucleic Acids Res. 2025;53(16):gkaf806.
Comment 5: The author's description in the experimental methods section is missing: the extraction and quantification methods of cfDNA in the article are not described in detail, and there is a lack of information on the use of reagent kits, measurement results, and other related information.
Response 5: We thank the reviewer for this helpful remark. We agree that the description of cfDNA extraction and quantification methods in the original submission was insufficient. We have now added more detailed information in the Methods section [line 137-143], including the specific kits and reagents used, the procedures for cfDNA isolation, and the approaches used for quantification.
Comment 6: Some figure legends are overly long and contain explanatory information, operation methods, etc. related to the figures. The explanation of the legend should be a simple description
Response 6: We appreciate the reviewer’s suggestion. We have revised and shortened the legends for Figures 2, 3, and 4 so that they now provide concise descriptions of the figures.
Comment 7: The author's article lacks a benchmark for cfDNA detection: No comparison with established methods (ddPCR, NGS with error correction) was performed. Without this, it is unclear whether the CRISPR method offers a real advantage. At the same time, no comparison was made with methods from other literature to highlight the advantages of our method in terms of accuracy, detection time, and simplicity of operation.
Response 7: We appreciate the reviewer’s important comment regarding benchmarking against established cfDNA detection methods. Indeed, highly sensitive approaches such as droplet digital PCR (ddPCR) and next-generation sequencing with molecular barcoding/error correction (NGS-EC) have been widely reported. For example, Oxnard et al. demonstrated that ddPCR could reliably detect EGFR mutations at an allele frequency down to ~0.01% [4, line 335-338], and Newman et al. showed that CAPP-Seq with error suppression enabled detection at ~0.004% [5, line 335-338] . In comparison, our mismatch-engineered Cas12a assay achieved a detection sensitivity of approximately 0.001%, which is comparable or better than the sensitivity range of leading cfDNA detection methods. Moreover, as our CRISPR-based approach selectively enriches mutant alleles, the it can be combined with aforementioned detection methods including ddPCR and CAPP-Seq to offer even higher detection sensitivity for low-frequency mutant alleles. Therefore, we envision that mismatch-engineered CRISPR detection is not necessarily a replacement for existing gold-standard methods but rather a complementary tool. When combined with established techniques—for example, initial screening by CRISPR assay followed by orthogonal confirmation with ddPCR or NGS—this strategy could provide both efficiency and reliability in clinical cfDNA analysis.
[4] Oxnard GR, et al. Noninvasive detection of response and resistance in EGFR-mutant lung cancer using quantitative next-generation genotyping of cell-free plasma DNA. Clin Cancer Res. 2014;20(6):1996–2004.
[5] Newman AM, et al. An ultrasensitive method for quantitating circulating tumor DNA with broad patient coverage. Nat Med. 2014;20(5):548–554.
Comment 8: The statement in the conclusion that it can be applied in treatment is speculative and not closely related to this article. There is no direct evidence to support the author's method in relation to disease treatment in this article.
Response 8: We appreciate the reviewer’s comment and agree that the statement in the conclusion regarding potential therapeutic application was speculative and not directly supported by our data. We have therefore removed this statement from the revised manuscript to ensure that the conclusions remain closely aligned with the presented evidence [Original manuscript version lines 408-410]. Specifically, the sentence “In addition to the diagnostic field, we expected that the distinction between WT and mutated DNA can be applied to various fields such as cancer therapy through pathogenic allele-specific genome editing” has been deleted from the conclusion.
Comment 9: The original image of the gel image provided by the author does not seem to be complete. Please provide complete and uncut image information.
Response 9: We thank the reviewer for pointing this out. The original gel images we provided contained some unnecessary background information, which may have made them appear incomplete. To address this, we have recompiled and submitted full, uncut gel images that retain all relevant lanes and markers while removing extraneous elements.
Comment 10: There are some grammar errors and inappropriate wording in the manuscript, which require language polishing.
Response 10: We thank the reviewer for this helpful remark. We have carefully proofread the entire manuscript and corrected grammatical errors and inappropriate wording.
Reviewer 2 Report
Comments and Suggestions for Authors
- This LbCas12a based diagnostic method doesn’t provide the allele frequency of the EGFR mutations, and clonality. Only the presence of a targeted mutation can be confirmed from the patient’s cfDNA. But the therapeutic strategy and dose for NSCLC treatment highly dependent on mutation AF and clonality. For which, the biopsy based genetic diagnosis is necessary. Authors are encouraged to discuss this matter elaborately in the ‘Discussion’ section.
- In figure 4.(b), authors are encouraged to include the control lanes of normal DNA and deleted DNA without the treatment of LbCas12a and WT crRNA.
- The details about the next-gen sequencing are encouraged to be provided in terms of coverage depths, paired-end or single-end sequencing, quality cut-off values and other relevant details.
- Authors need to show the percentage efficiency of the in-vitro cleavage. Is it 100% efficient? In the next-gen sequencing, what is percentage of base calling that shows the wild-type variant?
- None of the original gel images that authors submitted can be opened. It shows as ‘potential harmful file’ during opening. Authors are encouraged to resubmit those original gel images processed through some authentic image processors to avoid such a conflict with institutional firewall system.
Author Response
Comment 1: This LbCas12a based diagnostic method doesn’t provide the allele frequency of the EGFR mutations, and clonality. Only the presence of a targeted mutation can be confirmed from the patient’s cfDNA. But the therapeutic strategy and dose for NSCLC treatment highly dependent on mutation AF and clonality. For which, the biopsy based genetic diagnosis is necessary. Authors are encouraged to discuss this matter elaborately in the ‘Discussion’ section.
Response 1: We thank the reviewer for this valuable comment. We fully agree that allele frequency (AF) and clonality are critical determinants for therapeutic decision-making in NSCLC, as treatment strategy and dosing often depend on the quantitative burden of EGFR mutations. We acknowledge that our current LbCas12a-based assay provides information on the presence/absence of the mutation, which might be insufficient for assessment of AF or clonality. This might be an inherent limitation of the method in its current form. At the same time, the primary goal of our study was to establish the feasibility of mismatch-engineered Cas12a as a rapid and simple diagnostic tool for cfDNA, focusing on improving single-nucleotide discrimination and sensitivity. We see this approach as complementary rather than substitutive to biopsy-based genetic diagnosis and quantitative cfDNA assays such as ddPCR or NGS with error correction. In clinical practice, we anticipate that our CRISPR-based detection could serve as a rapid pre-screening tool to identify mutation-positive patients, followed by quantitative assessment through established methods to guide therapeutic strategy. We have revised the Discussion section [lines 393–404] to elaborate on this aspect and to note that future development may integrate Cas12a-based detection with quantitative platforms, thereby combining the speed and simplicity of CRISPR diagnostics with the quantitative power of established technologies.
Comment 2: In figure 4.(b), authors are encouraged to include the control lanes of normal DNA and deleted DNA without the treatment of LbCas12a and WT crRNA.
Response 2: We thank the reviewer for this helpful suggestion. As recommended, we repeated the experiment for Figure 4(b) and included the control lanes of normal DNA and deleted DNA without treatment of LbCas12a and WT crRNA. The figure has been revised accordingly, and the corresponding original gel image files have been provided in the revised submission.
Comment 3: The details about the next-gen sequencing are encouraged to be provided in terms of coverage depths, paired-end or single-end sequencing, quality cut-off values and other relevant details
Response 3: We thank the reviewer for this helpful suggestion. We agree that more detailed information about the next-generation sequencing workflow would improve clarity and reproducibility. Accordingly, we have revised the Methods section (Section 2.5, line 153-158) to include these details. Specifically, we now describe the preparation of target-specific libraries through nested PCR with barcode attachment, sequencing on the Illumina iSeq 100 platform using paired-end 150 bp reads, approximate coverage depth (~10,000×), and the quality filtering parameters (with >90% of bases typically exceeding Q30).
Comment 4: Authors need to show the percentage efficiency of the in-vitro cleavage. Is it 100% efficient? In the next-gen sequencing, what is percentage of base calling that shows the wild-type variant?
Response 4: We thank the reviewer for pointing out the importance of quantifying cleavage efficiency. Based on band-intensity quantification using ImageJ, the cleavage efficiencies were 36.9% (WT) and 36.3% (L858R) in Figure 2(a); 44.7% (WT) and 1.6% (L858R) in Figure 2(d); and 69.7% (WT) and 10.5% (Exon 19-deleted) in Figure 4(b). We conducted NGS after the cleavage, and indicated the detected mutation rates in the manuscript (Figure 3, 4).
Comment 5: None of the original gel images that authors submitted can be opened. It shows as ‘potential harmful file’ during opening. Authors are encouraged to resubmit those original gel images processed through some authentic image processors to avoid such a conflict with institutional firewall system.
Response 5: We apologize for the inconvenience caused by the corrupted original gel image files. We have reprocessed the original gel images and resubmitted them in a compatible format.
Round 2
Reviewer 1 Report
Comments and Suggestions for Authors
Based on the reviewers' comments, the manuscript has been carefully revised and greatly improved. It could fulfill the standards of the journal, and thus I recommend the acceptance in the journal.